# Effect of a-SiC_x_N_y_:H Encapsulation on the Stability and Photoluminescence Property of CsPbBr_3_ Quantum Dots

**DOI:** 10.3390/nano13071228

**Published:** 2023-03-30

**Authors:** Zewen Lin, Zhenxu Lin, Yanqing Guo, Haixia Wu, Jie Song, Yi Zhang, Wenxing Zhang, Hongliang Li, Dejian Hou, Rui Huang

**Affiliations:** 1School of Materials Science and Engineering, Hanshan Normal University, Chaozhou 521041, China; 2636@hstc.edu.cn (Z.L.);; 2National Laboratory of Solid State Microstructures/School of Electronics Science and Engineering/Collaborative Innovation Center of Advanced Microstructures, Nanjing University, Nanjing 210093, China

**Keywords:** a-SiC_x_N_y_:H encapsulation, CsPbBr_3_ QDs, stability, photoluminescence

## Abstract

The effect of a-SiC_x_N_y_:H encapsulation layers, which are prepared using the very-high-frequency plasma-enhanced chemical vapor deposition (VHF-PECVD) technique with SiH_4_, CH_4_, and NH_3_ as the precursors, on the stability and photoluminescence of CsPbBr_3_ quantum dots (QDs) were investigated in this study. The results show that a-SiCxNy:H encapsulation layers containing a high N content of approximately 50% cause severe PL degradation of CsPbBr_3_ QDs. However, by reducing the N content in the a-SiCxNy:H layer, the PL degradation of CsPbBr_3_ QDs can be significantly minimized. As the N content decreases from around 50% to 26%, the dominant phase in the a-SiCxNy:H layer changes from SiNx to SiCxNy. This transition preserves the inherent PL characteristics of CsPbBr_3_ QDs, while also providing them with long-term stability when exposed to air, high temperatures (205 °C), and UV illumination for over 600 days. This method provided an effective and practical approach to enhance the stability and PL characteristics of CsPbBr_3_ QD thin films, thus holding potential for future developments in optoelectronic devices.

## 1. Introduction

Recent studies have demonstrated that inorganic cesium lead halide perovskite (CsPbX_3_, Cl, Br, and I) quantum dots (QDs) have the potential to be used in optoelectronic applications, such as light-emitting diodes (LEDs) and high-definition displays, due to their high quantum yields (QYs), ultralow-voltage operation, and ultra-narrow room-temperature emission [1,2,3,4,5,6,7,8,9]. However, for CsPbX_3_ quantum dots, their crystal structure is inherently unstable, making them vulnerable to ion migration and decomposition in high-temperature, light, or humid conditions [10,11,12,13,14,15]. It has been found that oxygen and light play significant roles in the degradation of CsPbBr_3_ quantum dots. They facilitate the maturation and growth of these quantum dots, which leads to a decrease in fluorescence quantum efficiency [14,15]. Under oxygen and light conditions, water vapor also acts as an ion transport channel, thus accelerating the degradation of CsPbBr_3_ quantum dots [15]. Obviously, their poor stability when exposed to moist air, UV radiation, and high temperatures has been a barrier to their practical applications [10,11,12,13,14]. To overcome this, various strategies, such as doping engineering, surface engineering, and encapsulating engineering have been employed in an attempt to improve their stability [10,16,17,18,19,20,21,22,23]. For example, Zou et al. [16] utilized the substitution of Mn^2+^ to effectively stabilize perovskite lattices of CsPbX_3_ QDs even under ambient air conditions with temperatures as high as 200 °C. Pan et al. [10] developed a postsynthesis passivation process for CsPbI_3_ NCs by using a bidentate ligand, namely 2,2′-iminodibenzoic acid. This approach greatly enhanced the stability of red CsPbI_3_ quantum dots, resulting in improved LED device performance. Kim et al. [17] reported highly efficient and stable CsPbBr_3_ QDs, which retained more than 90% of the initial PLQY after 120 days of environmental storage by in situ surface reconstruction of CsPbBr_3_-Cs_4_PbBr_6_ nanocrystals. In comparison to doping engineering and surface engineering, CsPbX_3_ quantum dot composites produced through encapsulating engineering display higher stability [19,20,21,22,23]. Encapsulating engineering involves the use of an inert material layer to cover perovskite quantum dots, serving as a barrier against gas and ion diffusion and limiting the structure of the quantum dots. This method reduces the impact of air, light, water, and heat on the quantum dots, leading to improved stability. Moreover, the composite structure not only enhances the stability of perovskite quantum dots but also effectively passivates their surfaces, reducing surface defect states and improving photoluminescence (PL) efficiency. Therefore, utilizing encapsulating engineering for composite materials is a promising approach for enhancing the performance of CsPbX_3_ quantum dots [19,20,21,22,23]. For example, CsPbBr_3_/SiO_2_ Janus nanocrystal films displayed higher photostability with only a slight drop (2%) in the PL intensity after nine hours of UV illumination [19]. Loiudice et al. [20] successfully prepared AlO_x_ on CsPbX_3_ QD thin films using a low-temperature atomic layer deposition process, which improved their stability at high temperatures and under light exposure for hours. Despite the advances in encapsulation, the development of CsPbX_3_ QDs with both high efficiency and excellent stability for practical applications remains challenging due to their sensitivity to the environment and the chemicals used in the encapsulation process, which may attack CsPbX_3_ QDs and produce more defects, causing a marked deterioration of their PL intensity [14,22,23]. In our previous work [22], we developed a glow discharge plasma process combined with in situ real-time monitoring diagnosis to enhance the stability of CsPbBr_3_ QDs, and demonstrated that an a-SiN_x_:H encapsulating layer could significantly enhance the stability of CsPbBr_3_ QDs under air exposure, UV illumination, and thermal treatment. However, the PL intensity was drastically reduced by 60% after being encapsulated by the a-SiN_x_:H. Recently, to preserve the intrinsic photoluminescence (PL) characteristics of CsPbBr_3_ QDs, we developed a damage-free plasma based encapsulation technique with real-time in situ diagnosis for CsPbBr_3_ QD films. Our research revealed that the CH_4_/SiH_4_ plasma had negligible destructive effects on CsPbBr_3_ QDs. Using low-temperature plasma-enhanced chemical vapor deposition, we fabricated a-SiC_x_:H films that safeguarded the CsPbBr_3_ QDs from surface damage during encapsulation, sustaining the PL efficiency. However, despite our efforts, the CsPbBr_3_ QDs encapsulated by a-SiC_x_:H still degraded after two months [23].

In this work, the effect of a-SiC_x_N_y_:H encapsulation layers on the stability and photoluminescence (PL) of CsPbBr_3_ QDs was investigated. These layers were prepared using a very-high-frequency plasma-enhanced chemical vapor deposition (VHF-PECVD) technique with SiH_4_, CH_4_, and NH_3_ as precursors. It is found that a-SiC_x_N_y_:H encapsulation layers with a high N content of ~50% cause a serious PL degradation of CsPbBr_3_ QDs. However, by reducing the N content in the a-SiC_x_N_y_:H layer, the PL degradation of CsPbBr_3_ QDs can be significantly minimized. As the N content decreases from around 50% to 26%, the dominant phase in the a-SiC_x_N_y_:H layer changes from SiN_x_ to SiC_x_N_y_, which not only makes CsPbBr_3_ QDs retain their inherent PL characteristics but also endows CsPbBr_3_ QDs with long-term stability when exposed to air, at a high temperature (205 °C), and under UV illumination for more than 600 days.

## 2. Materials and Methods

A very-high-frequency plasma-enhanced chemical vapor deposition (VHF-PECVD) technique was employed to prepare a-SiC_x_N_y_:H/CsPbBr_3_ QDs/a-SiC_x_N_y_:H nanocomposite films with a sandwiched structure. The a-SiC_x_N_y_:H sublayer had a thickness of 15 nm and was firstly fabricated on silicon and quartz substrates using a mixture of SiH_4_, CH_4_, and NH_3_. The flow rates of SiH_4_ and CH_4_ were set at 2.5 SCCM (short for ‘standard cubic centimeters per minute’) and 6 SCCM, respectively, while the NH_3_ flow rate was different from 0 to 15 SCCM. The RF power, deposition pressure, and substrate temperature were maintained at 20 W, 20 Pa, and 150 °C, respectively. The 0.25 mg/mL CsPbBr_3_ QDs solution was spin-coated on the substrates at a speed of 6000 rpm for 30 s, followed by the deposition of a 15 nm thick a-SiC_x_N_y_:H film. The CsPbBr_3_ QDs were synthesized according to the procedures described by Protesescu et al. [1] To synthesize the CsPbBr_3_ quantum dot, 5 mL of ODE and 0.188 mmol (0.069 g, ABCR, 98%) PbBr_2_ were loaded into a 25 mL 3-neck flask and dried under vacuum at 120 °C for an hour. Then, 0.5 mL of dried oleylamine (OLA, Acros 80–90%) and 0.5 mL of dried OA were injected at 120 °C under a nitrogen atmosphere. Once the PbBr_2_ salt was completely solubilized, the temperature was raised to 140–200 °C, and a Cs-oleate solution (0.4 mL, 0.125 M in ODE) was quickly injected. Five seconds later, the reaction mixture was cooled using an ice-water bath. The structure and composition of the CsPbBr_3_ QDs/a-SiC_x_N_y_:H nanocomposite films were characterized using a Philips XL30 scanning electron microscope (SEM) and QUANTAX energy-dispersive X-ray spectroscope (EDS). The concentrations of Si, N, and C in the a-SiCxNy:H film were further determined by X-ray photoelectron spectroscopy (XPS). Optical absorption spectra were obtained via a Shimadzu UV-3600 spectrophotometer on quartz samples and used to estimate the optical bandgaps of a-SiC_x_N_y_:H films. For the acquisition of PL spectra, the Jobin Yvon FluoroLog-3 spectrophotometer was utilized, which is equipped with a 450 W continuous Xe lamp. Meanwhile, PL decay curves were recorded using an Edinburgh FLS980 spectrometer at room temperature. The temperature-dependent PL spectra were subsequently obtained by using a Raman Spectrometer that was equipped with a Linkam THMS 600 (Horiba LabRAM HR Evolution) from 25 to 205 °C.

## 3. Results and Discussion

Figure 1a shows the SEM image obtained from the CsPbBr_3_ QDs/a-SiC_x_N_y_:H nanocomposite thin film, which indicates that the thin film was uniform. The EDS elemental maps displayed in Figure 1b,c reveal that the Cs, Pb, Br, Si, N, and C elements were well distributed. This confirms that CsPbBr_3_ QDs are covered uniformly by an a-SiC_x_N_y_:H layer.

Figure 2 presents the PL from the CsPbBr_3_ QDs before and after being encapsulated by a-SiC_x_N_y_:H layers, which were prepared with various NH_3_ flow rates, respectively. As shown in Figure 2a, the PL intensity dropped remarkably by more than 40% after the CsPbBr_3_ QDs were covered by the a-SiC_x_N_y_:H encapsulating layer prepared at a high NH_3_ flow rate of 15 SCCM. With the decrease in NH_3_ flow rate, the decline of PL from the CsPbBr_3_ QDs was gradually reduced, as shown in Figure 2b–d. It was found that PL intensity from the CsPbBr_3_ QDs remained nearly unchanged after being covered by the a-SiC_x_N_y_:H layer prepared at a low NH_3_ flow rate of 5 SCCM. Obviously, PL intensity from CsPbBr_3_ QDs is closely related to the NH_3_ flow rate. As is demonstrated in our previous work [22], NH_3_ can produce N-related reactive species in the glow discharge plasma through the collision of electrons and NH_3_ molecules. Due to the high sensitivity of CsPbX_3_ QDs to the environment, N-related reactive species interact with the atoms on the CsPbX_3_ QDs surface and facilitatively create surface defects, leading to the degradation of the CsPbBr_3_ QDs in the encapsulation process. The effect becomes more severe with increasing NH_3_ flow rate. Therefore, the significant degradation of the CsPbBr_3_ QDs after encapsulation by a-SiC_x_N_y_:H mainly stemmed from the high NH_3_ flow rate used in the fabrication process. Figure 2c shows that a low NH_3_ flow rate of 5 SCCM had little detrimental effect on the CsPbBr_3_ QDs, suggesting that a low NH_3_ flow rate is more suitable for forming a-SiC_x_N_y_:H encapsulating layers on the CsPbBr_3_ QDs than a high NH_3_ flow rate.

From Figure 2d, it can be observed that without the addition of ammonia, the luminescence intensity of the CsPbBr_3_ QDs encapsulated with a-SiC_x_:H was weaker than that of the uncoated CsPbBr_3_ QDs. Additionally, the PL peak position of the SiC_x_-encapsulated CsPbBr_3_ QDs exhibited a redshift, and the full width at half maxima (FWHM) of the PL were narrower as compared to the uncoated CsPbBr_3_ QDs. This phenomenon could possibly be attributed to the self-absorption effect of the a-SiC_x_:H coating on the CsPbBr_3_ QDs. To further test this hypothesis, the transmission spectra of a-SiC_x_N_y_:H films prepared using different NH_3_ flows were measured, as depicted in Figure 3. As the NH_3_ flow rate decreased from 15 SCCM to 0 SCCM, the absorption edge of the transmission spectrum progressively shifted towards the longer wavelength region. This suggests that the optical band gap of the film gradually decreased with the decrease of NH_3_ flow rate. According to the formula [24]: αd = −ln T, where T is the transmittance and d is the film thickness, the absorption coefficient α of the film can be obtained. Thus, the Eopt can be calculated according to the Tauc equation (αhν)^1/2^ = Β (hν − Εopt) [25], where α is the absorption coefficient and B is a constant. From the inset of Figure 3, it is evident that the E_opt_ of the a-SiC_x_N_y_:H decreased from 5.20 to 2.55 eV when the NH_3_ flow rate was decreased from 15 to 0 SCCM. This variation indicates an evolution in the band structure of the a-SiC_x_N_y_:H attributed to the reduced N concentration as shown in the following Figure 5a. It is worth noting that the band gap value of the a-SiC_x_:H film (2.55 eV) coincides with the short wave region of the luminescent peak of the CsPbBr_3_ QDs at an NH_3_ flow rate of 0, thereby explaining the PL decay due to the self-absorption effect of the a-SiC_x_:H coating. It is therefore clear that the incorporation of N in the a-SiC_x_:H coating plays a crucial role in increasing the optical band gap and avoiding PL decay caused by the self-absorption effect of the coating.

To understand the PL characteristics, PL decay curves of the CsPbBr_3_ QDs/a-SiC_x_N_y_:H nanocomposite thin films prepared with different NH_3_ flow rates were measured under an excitation wavelength of 375 nm (375 nm, 70 ps excitation pulses LASER), as illustrated in Figure 4a. The PL decay curves were well fitted with a biexponential decay function [26,27,28]:(1)$$I(t)=I_{0}+A_{1}exp(\frac{-t}{\tau_{1}})+A_{2}exp(\frac{-t}{\tau_{2}})$$
where *I*_0_ is the background level; *τ*_1_, *τ*_2_ are the lifetime of each exponential decay component, and *A*_1,_
*A*_2_ are the corresponding amplitudes, respectively. Thus, the average lifetime τ can be estimated as follows [27]:$\tau =(A_{1}*\tau_{1}^{2}+A_{2}*\tau_{2}^{2})/(A_{1}*\tau_{1}+A_{2}*\tau_{2})$.

Figure 4b shows that the average lifetime (*τ*) increased gradually from 3.5 ns to 8.5 ns with the NH_3_ flow rate decreasing from 15 SCCM to 5 SCCM. The increase in the average lifetime usually means that nonradiative decay is somewhat suppressed and more excitons tend to recombine along radiative paths [29], which is in good agreement with the improved PL shown in Figure 2. Therefore, the a-SiC_x_N_y_:H layers prepared with a low NH_3_ flow rate are believed to effectively reduce the surface defect states of CsPbBr_3_ QDs and thus achieve efficient photoluminescence. It seems that the PL intensity from CsPbBr_3_ QDs is closely associated with the N content in the a-SiC_x_N_y_:H coatings.

The XPS spectra taken from the a-SiC_x_N_y_:H encapsulating layers were examined and the Si, N, and C contents in the films were estimated, as shown in Figure 5a. One can see that at the NH_3_ flow rate of 15 SCCM, the contents of Si, N, and C in the a-SiC_x_N_y_:H encapsulating layer were around 48%, 43%, and 9%, respectively. By decreasing the NH_3_ flow rate down to 5 SCCM, the N content sharply decreased to ~30%, while the Si and C contents rose to ~55% and ~15%, respectively. Without the addition of NH_3_, the Si and C contents of the encapsulating layer were around 60% and 40%, respectively. To gain further insight into the a-SiC_x_N_y_:H encapsulating layers, we analyzed the Si 2p core level spectra, which is the symbol of the coexistence of different ionic states of Si atoms [30,31,32], as shown in Figure 5b. It is noted that the binding energy peaks are typically composed of various Si phases, namely SiN_x_, SiC_x_N_y_, and SiC_x_. At a high NH_3_ flow rate of 15 SCCM, the a-SiC_x_N_y_:H encapsulating layer was dominated by the SiN_x_ phase, which was attributed to the high content of nitrogen as illustrated in Figure 2a. With the decrease of nitrogen flow rate from 15 to 5 SCCM, the dominant phase in the encapsulating layer changed from the SiN_x_ phase into the SiC_x_N_y_ phase. Without the addition of NH_3_, encapsulating layers feature the SiC_x_ phase. Nitrogen plays a key role in the chemical bond reconstruction and the phase transformation in the a-SiC_x_N_y_:H encapsulating layer. Combined with the analysis of PL characteristics, it was found that the encapsulation layer with the SiC_x_N_y_ phase was more beneficial to obtaining efficient PL than that with the SiN_x_ phase. As is illustrated in Figure 5b, the phase structure of nitride-rich a-SiC_x_N_y_:H film was dominated by the amorphous SiN_x_ phase, for which more N-active groups were generated in the preparation process. Excessive N-active precursors enhance the corrosion effect on the surface of CsPbBr_3_ QDs, creating more surface defects, which in turn weaken the passivation effect of the encapsulation layer on CsPbBr_3_ QDs. Accordingly, the PL intensity decreased significantly by more than 40% after the CsPbBr_3_ QDs were covered by the encapsulating layer with the SiN_x_ phase. For the encapsulating layer with SiC_x_N_y_ phase, fewer N active groups were produced in the preparation process, which weakened the corrosion effect of active groups on the surface of CsPbBr_3_ QDs, helped to suppress the generation of non-radiative centers on the surface of CsPbBr_3_ QDs, and thus enhanced the passivation effect of the encapsulation layer on CsPbBr_3_ QDs. This is in line with the experimental phenomenon of the PL lifetime increasing with the decrease of nitrogen content. Therefore, the encapsulation layer with the SiC_x_N_y_ phase is more favorable to obtaining efficient PL than that with the SiN_x_ phase.

**Figure 5 nanomaterials-13-01228-f005:**
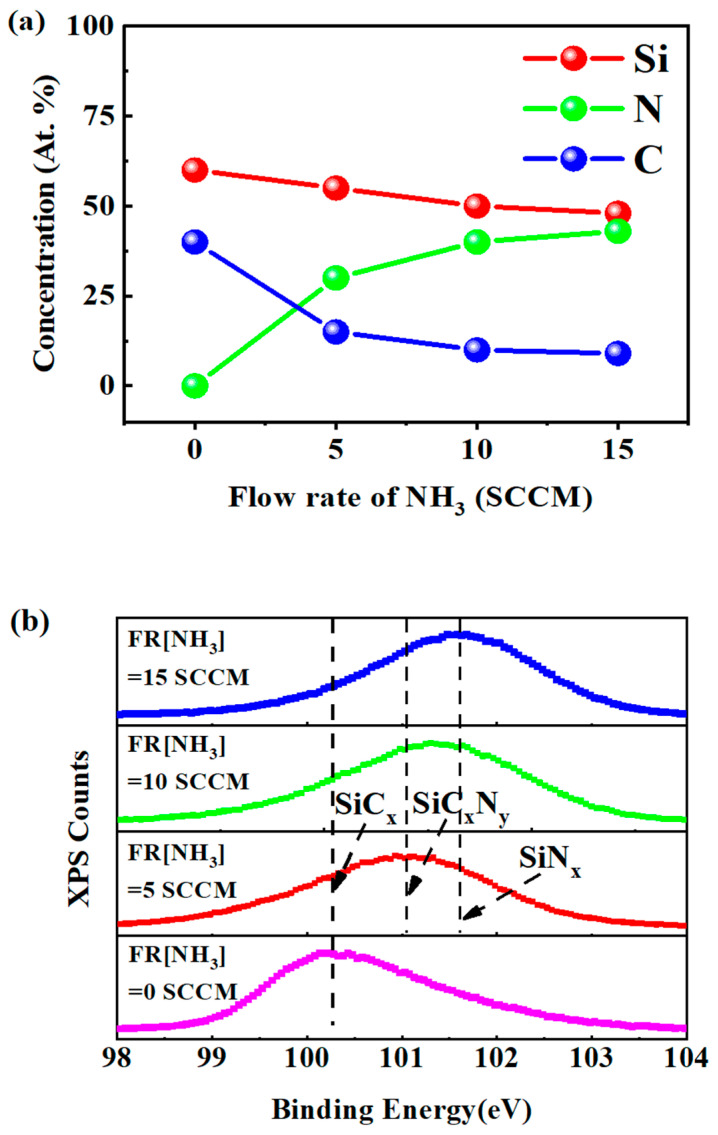
(**a**) The relative atomic concentration of Si, N, and C elements in a-SiC_x_N_y_:H encapsulating layers with different NH_3_ flow rates. (**b**) Experimental Si 2p spectra for a-SiC_x_N_y_:H encapsulating layers fabricated at different NH_3_ flow rates.

We assessed the stability of the CsPbBr_3_ QDs encapsulated by the layers with SiC_x_N_y_ phase under various unfavorable environments. Figure 6a shows the change of PL intensity from CsPbBr_3_ QDs/a-SiC_x_N_y_: H nanocomposites with the storage time in the air and under UV light illumination, respectively. After storing for more than 600 days in the air (humidity and temperature ranging from 40% to 70% and 15 to 30 °C, respectively), no significant PL intensity decline was observed from the CsPbBr_3_ QDs/a-SiC_x_N_y_:H nanocomposites. They initially increased gradually and then stabilized after tripling. The enhanced PL was ascribed from the light irradiation during the PL measurement, which is referred to as photoactivation [22,33]. Similarly, the PL from CsPbBr_3_ QDs/a-SiC_x_N_y_:H nanocomposites rapidly increased by more than 3 times after UV irradiation for 1 day. The enhanced PL remained stable during subsequent continuous UV illumination for at least 600 days. It is interesting that CsPbBr_3_ QDs/a-SiC_x_N_y_:H nanocomposites, which were illuminated by continuous UV for at least 600 days, still showed strong green emissions even at the high temperature of 205 °C, as shown in Figure 6b. On the other hand, it is worth noting that the CsPbBr_3_ QDs encapsulated by the layers with SiC_x_ phase showed an obvious decline in PL after undergoing continuous UV light illumination for 2 months [23]. Evidently, the encapsulation layer with the SiC_x_N_y_ phase is more beneficial for obtaining stable and efficient PL than that with the SiC_x_ phase.

To assess the thermal stability of CsPbBr_3_ QDs/a-SiC_x_N_y_:H nanocomposite thin films, we monitored the integrated PL intensity as a function of temperature during thermal cycling, as shown in Figure 7. With increasing temperature from 25 to 205 °C, the PL was rapidly quenched. However, PL thermal quenching could be retrieved after the cooling process. Furthermore, the PL peak showed reversible modulation during the heating and cooling cycles. Compared to the irreversible PL deterioration of the original CsPbBr_3_ QDs during thermal cycling [22], the thermal stability of CsPbBr_3_ QDs/a-SiC_x_N_y_:H nanocomposite thin films was significantly improved, which can be ascribed to the constraint exerted by the CsPbBr_3_ QDs/a-SiC_x_N_y_:H interface as demonstrated by our previous work [22]. These results indicate that the encapsulation layer with SiC_x_N_y_ phase not only provides nondestructive encapsulation for CsPbBr_3_ quantum dots to achieve efficient PL, but also serves as a protective layer to realize the long-term stability of CsPbBr_3_ QDs under harsh environment.

## 4. Conclusions

The effect of a-SiC_x_N_y_:H encapsulation on the stability and PL of CsPbBr_3_ quantum dots were demonstrated. We found that a-SiC_x_N_y_:H encapsulation layer with a high N content of ~50% resulted in a serious PL degradation of CsPbBr_3_ QDs. The PL degradation of CsPbBr_3_ QDs can be significantly reduced by decreasing the N content in the a-SiC_x_N_y_:H encapsulation layer. With the N content decreasing from ~50% to ~26%, the dominant phase in the a-SiC_x_N_y_:H encapsulating layers transforms from the SiN_x_ to the SiC_x_N_y_ states. The encapsulation layer with the SiC_x_N_y_ phase not only makes CsPbBr_3_ QDs retain their inherent PL properties, but also makes CsPbBr_3_ QDs have long-term stability when exposed to air, at a high temperature (205 °C), and under UV illumination for more than 600 days. Our results demonstrate an effective and practical approach to enhance the stability and PL characteristics of CsPbBr_3_ QD thin films, with implications for the future development of optoelectronic devices.

## Figures and Tables

**Figure 1 nanomaterials-13-01228-f001:**
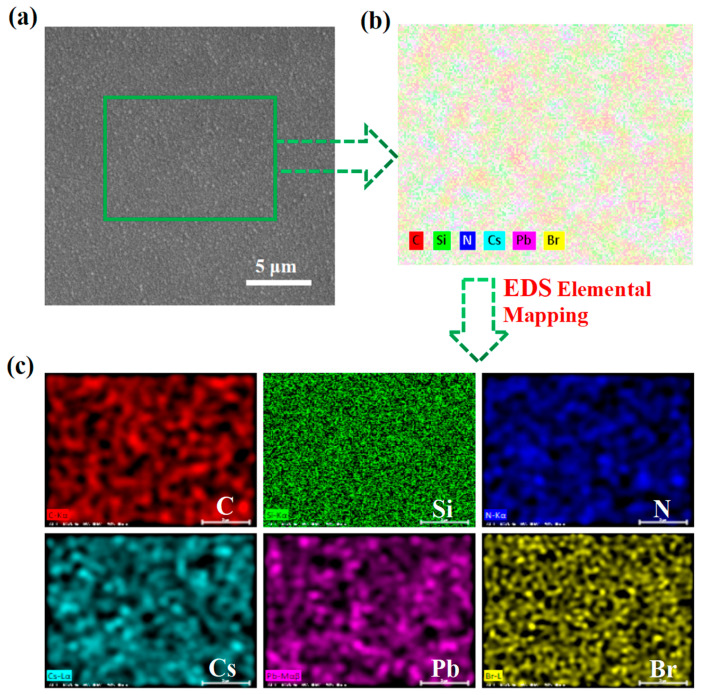
(**a**) SEM image of the CsPbBr_3_ QDs/a-SiC_x_N_y_:H nanocomposite thin film. (**b**) Elemental mapping of the CsPbBr_3_/a-SiC_x_N_y_:H nanocomposite thin film recorded by EDS. (**c**) EDS elemental mapping of Cs, Pb, Br, Si, N, and C in the thin film, respectively.

**Figure 2 nanomaterials-13-01228-f002:**
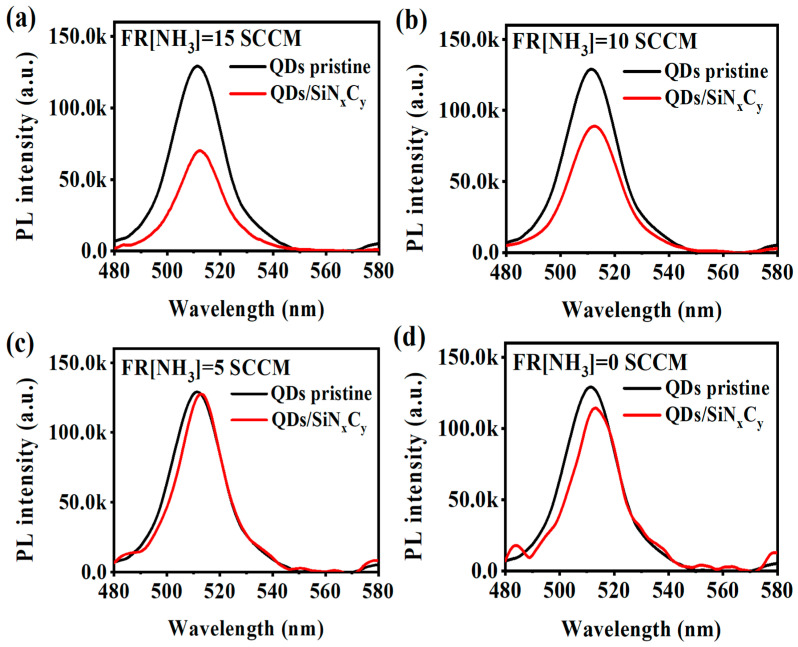
PL spectra acquired from the pristine CsPbBr_3_ QD film and CsPbBr_3_ QDs/a-SiC_x_N_y_:H nanocomposite thin films prepared with different NH_3_ flow rates: (**a**) 15 SCCM; (**b**) 10 SCCM; (**c**) 5 SCCM; (**d**) 0 SCCM, respectively.

**Figure 3 nanomaterials-13-01228-f003:**
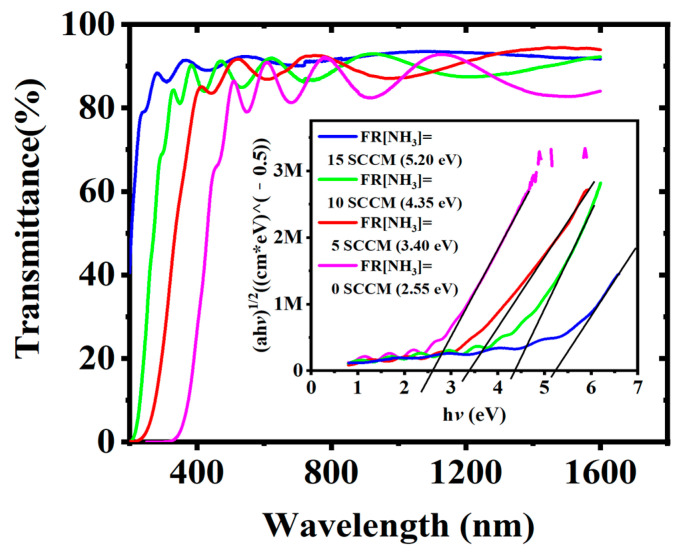
Transmission spectra of the a-SiC_x_N_y_:H films fabricated with different NH_3_ flow rates. Inset shows the Tauc plots of the corresponding a-SiC_x_N_y_:H films.

**Figure 4 nanomaterials-13-01228-f004:**
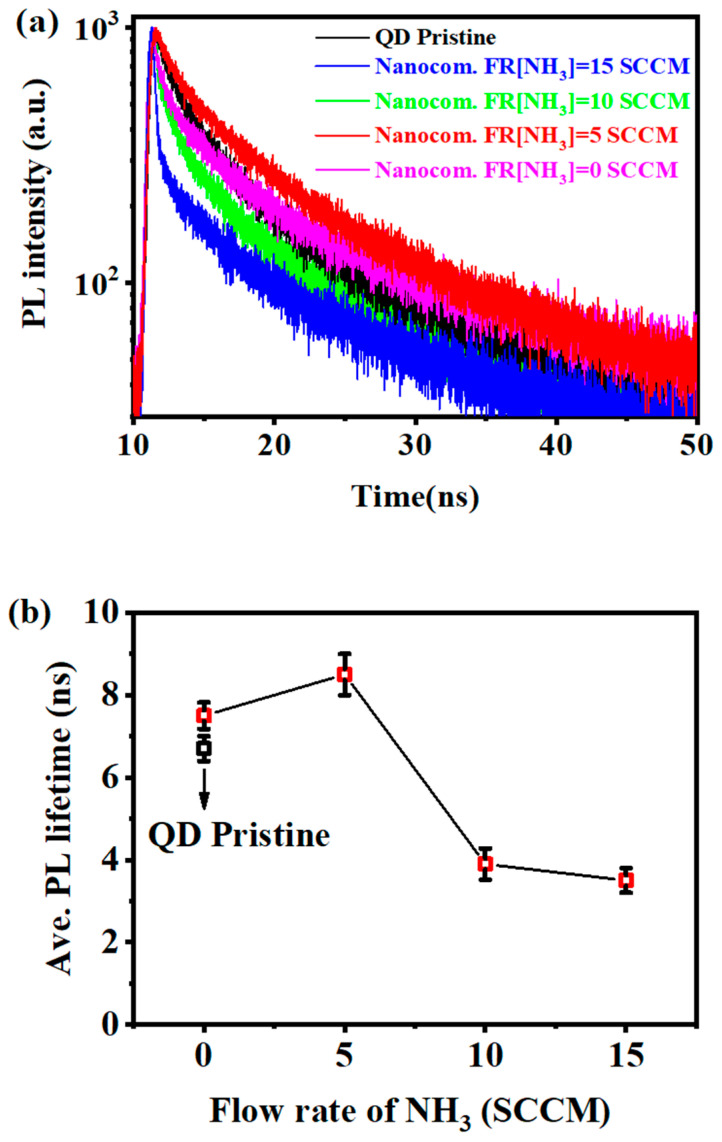
(**a**) PL decay traces and (**b**) lifetime of the pristine CsPbBr_3_ QDs thin film and CsPbBr_3_ QDs/a-SiC_x_N_y_:H nanocomposite thin films prepared with different NH_3_ flow rates.

**Figure 6 nanomaterials-13-01228-f006:**
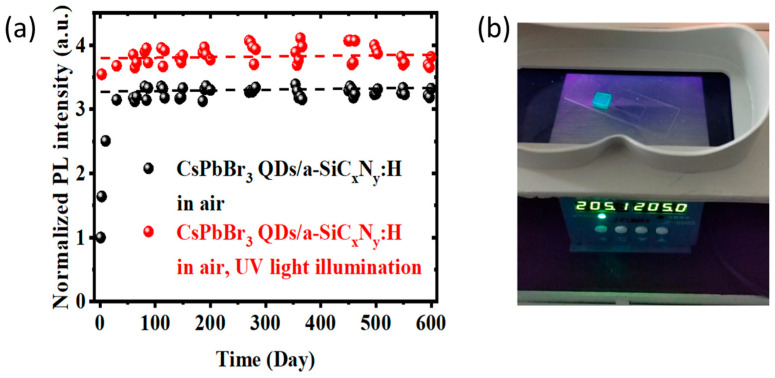
(**a**) PL intensity of the CsPbBr_3_ QDs encapsulated by the layers with SiC_x_N_y_ phase versus time under ambient conditions and continuous UV (365 nm, 8 W) illumination time. (**b**) Photographs taken during UV illumination (λ = 365 nm, 8 W) at a high temperature of 205 °C.

**Figure 7 nanomaterials-13-01228-f007:**
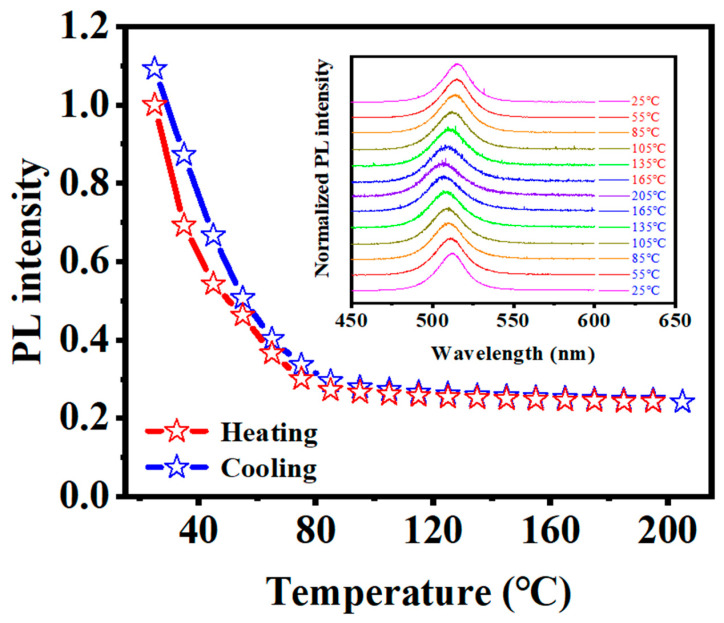
Heating and cooling cycling measurements of the CsPbBr_3_ QDs encapsulated by the layers with SiC_x_N_y_ phase at various temperatures. The inset presents the PL spectra vs. temperature.

## Data Availability

Data underlying the results presented in this paper are not publicly available at this time but may be obtained from the authors upon reasonable request.

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
