# Peer review of "Effect of a-SiCxNy:H Encapsulation on the Stability and Photoluminescence Property of CsPbBr3 Quantum Dots"

_nanomaterials, 2023, doi:10.3390/nano13071228_

Round 1

Reviewer 1 Report

Report

The authors presented work entitled “Effect of a-SiCxNy:H encapsulation on the stability and photo- 2 luminescence property of CsPbBr3 quantum dots”. In this work the effect of a-SiCxNy:H encapsulation layers, which are prepared by a very high frequency plasma enhanced chemical vapor deposition technique with SiH4, CH4 and NH3 as precursors, on the stability and photoluminescence of CsPbBr3 quantum dots have been considered. The authors confirm that a-SiCxNy:H encapsulation layers with a high N content of about 50% cause severe PL degradation of CsPbBr3 QDs. The PL degradation of CsPbBr3 QDs can be reduced by decreasing the N content in the a-SiCxNy:H encapsulation layer.

·       The authors must show the novelty of this work.

·       The authors mention that the structure proposed is stable. They must show this phenomenon.

·       The work presented by the authors contains a lot of ambiguity.

·       I suggest validation of this work.

In conclusion, this work needs improvement. 

Reviewer 2 Report

Please find my comments for authors in the attached file

Reviewer 3 Report

In the reviewed paper are presented the results of  investigation of the effect of a-SiCxNy:H encapsulation layers, which are prepared by VHF-PECVD technique with SiH4, CH4, and NH3 as the precursor, on stability and photoluminescence of CsPbBr3 quantum dots (QDs).

The method and technology of the sample's preparation (VHF-PECVD technique) are fully adequate to the task which has been declared. The experimental details are described clearly and fully.

The characterisation of the nanofilms was done using highly effective methods and equipment (scanning electron microscopy (SEM) with 2D elementary analysis and energy-dispersive X-ray spectroscopy (EDS),  X-ray photoelectron spectroscopy (XPS), Edinburgh FLS980 spectrometer equipped to PL detection and Raman Spectrometer equipped with a Linkam THMS 600).

As a result, the article contains results that are interesting from both scientific and practical points of view.

The factual material and its discussion are presented quite clearly and completely.

There are, from my point of view, some debatable points. 

For example, 1) the direct correlation by the authors of the PL intensity with the QD efficiency, or 2) the error in changing the PL intensity during comparative measurements before and after encapsulation is not discussed. However, these are secondary issues.

Summary, the article contains results that are interesting from both scientific and practical points of view.

I think that the results presented in the paper and the form they are discussed can be presented for publication in the form chosen by the authors, without modification.

Author Response

Thank you very much for your high evaluation and beneficial suggestions for our paper.

Round 2

Reviewer 1 Report

In conclusion, the authors responded to the majority of may remarks and suggestions. I would like to thank the authors for thier efforts which provided. 

Author Response

Thanks again for your positive evaluation and beneficial suggestions for our paper.

Reviewer 2 Report

see file
